# FASTER VISION MAMBA IS REBUILT IN MINUTES VIA MERGED TOKEN RE-TRAINING

## ABSTRACT

Vision Mamba has shown close to state of the art performance on computer vision tasks, drawing much interest in increasing it's efficiency. A promising approach is token reduction (that has been successfully implemented in ViTs). Pruning informative tokens in Mamba leads to a high loss of key knowledge and degraded performance. An alternative, of merging tokens preserves more information than pruning, also suffers for large compression ratios. Our key insight is that a quick round of retraining after token merging yeilds robust results across various compression ratios Empirically, pruned Vims only drop up to 0.9% accuracy on ImageNet-1K, recovered by our proposed framework R-MeeTo in our main evaluation. We show how simple and effective the fast recovery can be achieved at minute-level, in particular, a 35.9% accuracy spike over 3 epochs of training on Vim-Ti. Moreover, Vim-Ti/S/B are re-trained within 5/7/17 minutes, and Vim-S only drops 1.3% with 1.2× (up to 1.5 ×) speed up in inference.

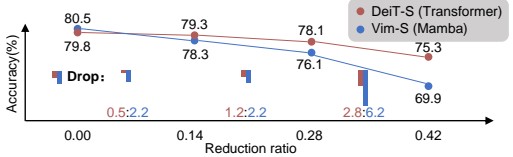

(a) Transformer v.s. Mamba: Mamba is more sensitive in pruning

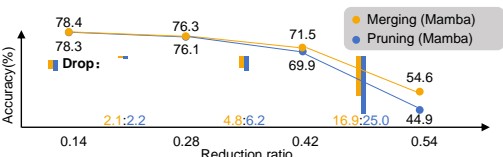

(b) Merging v.s. pruning: merging keep more information in Mamba.

Figure 1: Performance comparison *w.r.t.* reduction ratio: a) Transformer and Mamba in token pruning; b) Merging and pruning with Mamba. Transformer and Mamba are respectively DeiT-S (Touvron et al., 2021)/Vim-S (Zhu et al., 2024), tested on ImageNet-1K (Deng et al., 2009).

## 1 INTRODUCTION

Vision Mambas (e.g., Vim (Zhu et al., 2024)) have successfully introduced Mamba (Gu & Dao, 2023) into computer vision, achieving promising results (Zhu et al., 2024; Liu et al., 2024b; Pei et al., 2024; Yang et al., 2024a). The success of Vim and its follow-up works (Vims) is closely tied to the SSM (Gu & Dao, 2023) model's efficient sequence processing.

Token reduction is popular in model efficiency. The efficiency of DynamicViT (Rao et al., 2021) and its ilk (Meng et al., 2022; Liang et al., 2022; Chen et al., 2023) helps in conducting an effective Transformer with reduced tokens. EViT (Liang et al., 2022) identifies the informative tokens and simplifies the training process. AdaViT (Meng et al., 2022) shifts the view of computation reduction to attention heads and blocks, giving more flexibility to handling image tokens. Token pruning has yielded promising outcomes in ViTs (Rao et al., 2021; Liang et al., 2022; Pei et al., 2024), yet its efficiency in Vim remains unexplored. Token reduction performs well in input-agnostic Transformer (Bolya et al., 2023), while that on Mamba is unexplored. Therefore, comparative analyses of the performance between Transformer and Mamba are conducted in Fig. 1a. These present a challenge that the token pruning operation, designed on ViTs, performs less effectively on Vim.

Mamba's sequential dependency implies that token reduction varies from Transformer. A closer look at the difference between Transformer and Mamba: after a given time $t$, the token in Mamba contains more general knowledge than that in Transformer, due to the *enrichment effect of SSM: tokens are asymmetric in the amount of information they keep* (Theorem 1). As a result, the tokens at the end of a sequence share the most general knowledge. Pruning these informative tokens leads

to the high loss of general knowledge and performance drop. So pruning is not a good solution to make Mamba more efficient.

Token merging (Kong et al., 2022; Bolya et al., 2023) is an alternative solution for token reducing. It has demonstrated commendable performance in ViTs, and it preserves more token information than pruning. We present a performance comparison between pruning and merging under various reduction ratios, and the merging consistently outperforms pruning in Fig. 1b. Pruning directly removes tokens, resulting in loss of key knowledge. In contrast, merging preserves more key knowledge.

However, merging performance drops as the reduction ratio grows either, as shown in Fig. 1b. This suggests that training-free is not a good solution for maintaining key knowledge and performance in Mamba. In Tab. 1b, we observe that simply re-training the model with token merging enhances the performance of Mamba. We find that re-training effectively rebuilds the key knowledge in Mamba.

R-MeeTo (**R**e-training **Me**rged **To**ken) is therefore proposed. Its overall goal is to rebuild an effective pruned Mamba model with a faster inference speed. Empirically, R-MeeTo recovers token-reduced models' performance, leading only up to 0.9 drop on ImageNet-1K. We show that the recovery can be achieved at minutes-level. In particular, a 35.9 accuracy is regained over three epochs of training in only 4.2 minutes on Vim-Ti. The inference efficiency is up to 1.5x on RTX 4090s. We highlight the main contributions of this paper below:

- We hypothesize that the key knowledge loss in tokens mainly causes the token reduction's performance dropa, a view comprehensively from both theoretical and empirical research.
- A simple yet effective framework R-MeeTo, fast recovering key knowledge and performance, provides an direction for practical and industrial visual Mamba's efficiency.
- Our framework recovers the pruned models at minute-level, *e.g*, for Vim-Ti, R-MeeTo recovers 35.9% accuracy with only 8 minutes re-training on ImageNet-1K.

## 2 METHODOLOGY

### 2.1 PRELIMINARY: STATE SPACE MODELS

**Structured SSM.** Structured Sate Space Model (SSM) is a linear time-invariant system $w.r.t.$ time $t$, whose discrete form is shown as follows:

$$h_t = Ah_{t-1} + Bx_t, \quad y_t = Ch_t \tag{1}$$

where the state matrix $A$, the input matrix $B$, and the output matrix $C$ are three learnable parameters, $x_t$ and $y_t$ are respectively input and output, and $h_t$ is the hidden state at time $t$. $h_t$ and $h_{t-1}$ are simplified as $h$ and $h_-$ in following transfer equations and analyses. These diagonal plus low-rank structure are designed to compute sequence-to-sequence modules efficiently (Dao & Gu, 2024).

**Selective SSM.** In Mamba (Gu & Dao, 2023), proposed Selective SSM change the linear time-invariant system into non-linear time-variant system with a design of a discrete non-linear operator decided by $\mathbf{\Delta}_t$. $\mathbf{\Delta}_t$ is directly used as a gate to discrete $A$ and $B$, and further influence $C$, which change the system into the non-linear and time-variant one:

$$h_t = A_t h_{t-1} + B_t x_t, \quad y_t = C_t h_t \tag{2}$$

where $A_t$, $B_t$ and $C_t$ are dependent on $x_t$ and thus time-dependent version of the ones in Equ. 1. In this section, we use $X_t$ to represent the input of SSM $x_t$'s random variable, and $Y_t$ is output $y_t$'s random variable accordingly.

### 2.2 DISCUSSIONS

In this section, we propose explanations about the observations in Fig. 1. The analyses are based on the difference between the Attention Block and SSM from information transferring perspectives.

The **key knowledge** (*i.e.*, specific and general knowledge) is reduced by token reduction, and further the remnant tokens and their imbalance lead to performance drop. As shown in Fig. 2, the essential causes are: 1) a large amount of general knowledge is irreparably reduced; 2) specific knowledge keeping ratio is low and imbalanced. Further experiments in Fig. 3 support our theorem, if we shuffle token after reduction, only Mamba dropping, which means tokens in Transformer is not sensitive to its order of tokens' indexes. Moreover, token reduction's disruption to sequential dependency is one of the reasons for performance dropping, due to the knowledge embedded in the tokens' sequence.

**Takeaways.** The main conclusions are as follows:

*Mamba is more sensitive in pruning.* Due to the marginal enrichment effects proposed in Theorem 1, we show that it's a higher chance to prune tokens containing more general knowledge in Mamba, according to Corollary 2.

*Merging performs better in Mamba.* It's because merging keeps more key knowledge by token keeping and filtering with similarity, according to Corollary 3.

*Re-training is a simple and effective solution.* With most of the key knowledge still retained, the performance can simply recovered by re-training as shown in Tab. 1b.

## 2.3 DETAILED ANALYSES AND RELATED THEOREMS

In order to conduct fair comparison, the modules are assumed to be with the same information compression and extraction ability to get rid of the complicated impacts of the models' performance about scale of parameters, training tricks and others. $Y^{\mathbf{T}}$ and $Y^{\mathbf{M}}$ are the output signals of the Attention Block and SSM respectively.

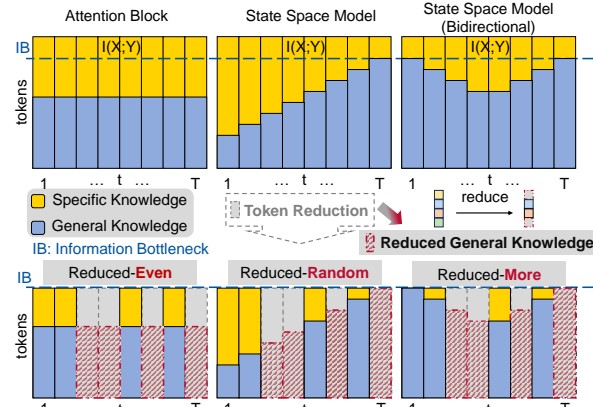

Figure 2: Analysis' sketch: Mamba is sensitive to token reduction from information perspective.

**Assumption 1.** *(Equal compressed information.) The output signals have the same entropy, both in general and in inference.*

$$\begin{cases} H(Y_{0:T}^{\mathbf{T}}) = H(Y_{0:T}^{\mathbf{M}}), \\ H(Y_{0:T}^{\mathbf{T}}|X_t) = H(Y_{0:T}^{\mathbf{M}}|X_t), \forall t \in [T]. \end{cases} \quad (3)$$

*where $H$ is the Shannon entropy, and $0 : T$ represents the discrete tokens' indexes from $0$ to $T-1$.*

**Assumption 2.** *(Equal amount of shared knowledge.) Given the same inputs or not, the same general knowledge amount share between $Y_{t:T}$ and $Y_{0:t}$ should be kept.*

$$\begin{cases} I(Y_{t:T}^{\mathbf{T}}; Y_{0:t}^{\mathbf{T}}) = I(Y_{t:T}^{\mathbf{M}}; Y_{0:t}^{\mathbf{M}}), \forall t \in [T], \\ I(Y_{t:T}^{\mathbf{T}}; Y_{0:t}^{\mathbf{T}}|X_t) = I(Y_{t:T}^{\mathbf{M}}; Y_{0:t}^{\mathbf{M}}|X_t), \forall t \in [T]. \end{cases} \quad \text{where $I$ is the mutual information.} \quad (4)$$

**Remark 1.** *Under Assumption 1, its equations allow us to focus on the equally expressive Attention and SSM modules, instead of the number of parameters, implementation tricks and detailed design involved. Assumption 2 guarantees that the amount of general knowledge is the same between Attention Blocks in Transformer and SSMs in Mamba, for any inputs $\{X_t\}_{t \in [T]}$.*

Thus, the effect of external factors to performance is ruled out with given assumptions, according to Remark 1, and we therefore study the inherent effects and differences between these modules' mechanisms as followings.

**Theorem 1.** *(Enrichment effect in Mamba.) Under Assumption 1 and Assumption 2, we have the following relationship between Attention Block and SSM.*

$$\begin{cases} I(Y_{t:T}^{\mathbf{M}}; X_t) \geq I(Y_{t:T}^{\mathbf{T}}; X_t), \forall t \in [T], \\ I(Y_{t:T}^{\mathbf{M}}; X_{t:T}) \geq I(Y_{t:T}^{\mathbf{T}}; X_{t:T}), \forall t \in [T]. \end{cases} \quad (5)$$

**Corollary 1.** *(Directions of SSM along time $t$ and reverse.) The direction of the inputs in SSM is only decided by time $t$. It means that in a reverse (backward) SSM, where time $t$ decrease from $T$ to $0$, we have a similar result as follows:*

$$\begin{cases} I(Y_{0:t+1}^{\mathbf{M}}; X_t) \geq I(Y_{0:t+1}^{\mathbf{T}}; X_t), \forall t \in [T], \\ I(Y_{0:t}^{\mathbf{M}}; X_{0:t}) \geq I(Y_{0:t}^{\mathbf{T}}; X_{0:t}), \forall t \in [T], \end{cases} \quad (6)$$

*which can be simply obtained by the same derivation as forward SSM and the commutative.*

**Remark 2.** *Theorem 1 shows that the tokens after time $t$ from a SSM block have more information about inputs at time $t$, comparing to what a equally expressive Attention Block have. Moreover, Corollary 1 shows that bidirectional SSM likewise has a knowledge enrichment effect, with the mutual information of the inputs $\{X_t\}_{t\in[T]}$ enriched in the direction of time $t$ change.*

**Proposition 1.** *(Marginal enrichment.) Conclusion of the 1st part: the tokens on both sides (i.e., those close to $0$ or $T$ in bidirectional SSM) concentrates more general knowledge. Moreover, the enrichment effect in Mamba is more serious than that in Transformer, according to Theorem 1.*

By information bottleneck theory (Tishby et al., 2000; Tishby & Zaslavsky, 2015), we view the inputs of each blocks $X_{0:T}$, the outputs of the Attention or SSM modules in the block $Y_{0:T}$, and ultimately the ideal output of the block $\hat{Y}_{0:T}$ (*e.g.*, the ground truth label for the last block, ideal tokens' representations for the next block), as a Markov chain: $\hat{Y} \rightarrow X \rightarrow Y$ (with simplified indexes). We have a objective of a well-trained model:

$$\min I(X;Y) - \beta I(\hat{Y};Y), \quad (7)$$

where $\beta$ is a given hyper-parameter.

According to Equ. 7, in a well-trained model, a large amount of the specific information about $X$ and $Y$ is

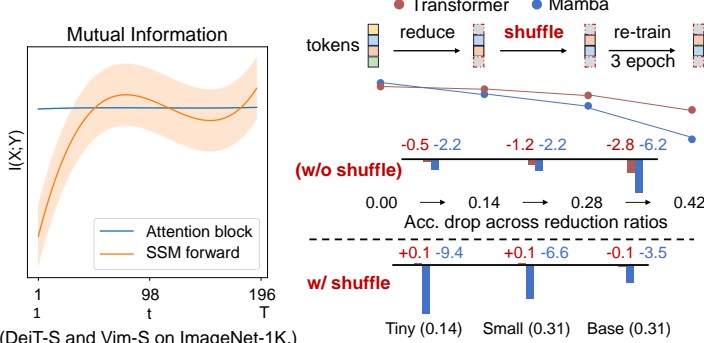

Figure 3: **Supporting facts.** 1) The empirical results of $I(X;Y)$, the mutual information between inputs $X$ and outputs $Y$. **Mamba is sensitive to token order.** 2) Only Mamba's performance drops if we further *Shuffle Tokens* before re-training. The Attention Block and SSM are measured by MINE (Belghazi et al., 2018) on the middle layers of DeiT-S and Vim-S (7-th/12 layers and the 14-th/24 layers respectively.) Experiments about i) token reduction are conducted with DeiT-S (Touvron et al., 2021) (Transformer) and Vim-S (Zhu et al., 2024) (Mamba) on ImageNet-1K (Deng et al., 2009). The reduction ratios in the experiment about ii) shuffled tokens are 0.14 for Vim-Ti and 0.31 for Vim-S/Vim-B (see Sec. 3.4 for more details about ablation). Shuffle strategy is odd-even shuffle: $[0,1,2,3]\rightarrow[0,2], [1,3]\rightarrow[0,2,1,3]$.

compressed, and more general knowledge is stored in $Y$. Further, Proposition 1 tell us that general knowledge is stored in tokens on both sides in Mamba. Thus, we have the following corollaries about the key knowledge, *i.e.* the specific and general ones, explaining the performance drop in Mamba's token reduction.

**Corollary 2.** *(Higher risk of token reduction in Mamba.) In Mamba, the further to the center the token is, the larger ratio of general knowledge it has, leading to the risk of general knowledge being removed. Meanwhile, the closer to the center the token is, the larger ratio of specific knowledge, leading to the risk of specific knowledge being removed. This doesn't happen in Transformers. Thus, pruning marginal or central tokens drops a much larger ratio of each key knowledge, and therefore drops performance due to the fact that performance is supported by both specific knowledge and general knowledge. Even in bidirectional SSMs, the tokens in both directions are distinct, simple additive introducing more noise and limited knowledge keeping.*

**Corollary 3.** *(Merging is better.) In merging, besides not directly deleting the token, filtering the general knowledge based on token similarity prevents the loss of all key knowledge in reduced tokens at once. Merging is thus better than pruning. However, similarity-based reduction do not prevent the loss of specific knowledge, and recovering this key knowledge is thus needed.*

### 2.4 METHOD

According to the aforementioned analyses, one of the main causes about Mamba's sensitivity and performance drop is the loss of key knowledge by token reduction. Therefore, in this section, we focus on the ways to effectively keep and recover the key knowledge in Mamba.

As discussed in Sec. 2.2 and Sec. 2.3, compared with pruning, merging remains the general knowledge in the reduction tokens by fusing similar tokens. As results in Tab. 1a, 1) merging keeps higher performance than pruning, especially in high reduction ratios. More given general knowledge re-

| reduction ratio | top-1 acc.(%) | | | reduction ratio | top-1 acc.(%) | | |
|---|---|---|---|---|---|---|---|
| | pruning | merging | $\Delta$ | | training-free | re-trained | $\Delta$ |
| 0.14 | 78.4 | 78.5 | 0.1↑ | 0.14 | 78.5 | 80.0 | 1.5↑ |
| 0.28 | 76.4 | 76.7 | 0.3↑ | 0.28 | 76.7 | 79.5 | 2.8↑ |
| 0.42 | 71.7 | 72.9 | 1.2↑ | 0.42 | 72.9 | 78.4 | 5.5↑ |
| 0.54 | 53.4 | 60.7 | 7.3↑ | 0.54 | 60.7 | 76.3 | 15.6↑ |

(a) **Merging vs. pruning.** Merging consistently outperforms pruning.

(b) **Re-training vs. training-free.** Re-training leads to higher performance.

Table 1: a) Comparison between token pruning and merging operations on the performance of Vim-S. $\Delta$ represents the difference in performance when using merging compared to pruning. Merging achieves higher top-1 accuracy (%) than pruning and retains more information from the unreduced tokens. b) Comparison between training-free and re-trained on the performance of token merged Vim-S. $\Delta$ is the difference between retraining or not. Re-training works effectively.

stored, 2) re-training is simple but effective to recover the performance in Mamba with limited specific knowledge, as shown in Tab. 1b.

Consistent with these facts empirically and theoretically, our proposal is that *both token merging and re-training should be combined* for key knowledge keeping and recovering. Our framework, R-MeeTo (**R**e-training **Me**rged **To**ken), is therefore simple, compatible, and effective simultaneously.

**R-MeeTo.** The whole process of our algorithm is shown in Algorithm 1 (in Appendix B). Merging and re-training are two main operations in our algorithm.

**R-MeeTo: Merging.** Every two blocks we perform a token merge operation. In each merge process, we pick the $r$ closest token pairs and add them into one token in each pair. The distance between tokens is measured by cosine between tokens' features as default.

**R-MeeTo: Re-training.** We minimize the standard cross-entropy loss on training set as default. As shown in Tab. 1b, performance increases dramatically after only 3 epochs, and thus we simply propose that re-training is a process with compatibility and efficiency for key knowledge recovering.

## 3 EXPERIMENT

### 3.1 SETTINGS

**Datasets and models.** We conduct all of our experiments on the ImageNet-1K (Deng et al., 2009) classification task and report top-1 accuracy (%). All images are augmented and resized to $224^2$ for evaluation. The baseline Mamba models comprise three variants of Vim (Zhu et al., 2024): Vim-Ti with 7 million parameters, Vim-S with 26 million parameters, and Vim-B with 98 million parameters. Following training techniques used in previous work (Touvron et al., 2021; Rao et al., 2021), all baseline models are initialized using pretrained weights (Zhu et al., 2024).

**Implementation details.** All experiments are conducted on a single machine equipped with 4 NVIDIA TESLA A100 40GB GPUs. During training, we use a batchsize of 128 with gradient accumulation performed over two steps, resulting in an effective total batchsize of 1024. Moreover, all models are trained with AdamW optimizer with a learning rate decaying from 2e-5 to 1e-6 using a cosine scheduler, and a weight decay of 5e-2. To ensure consistent FLOPs across R-MeeTo and other comparison methods, the token reduction ratio is set by default to 0.14 for Vim-Ti/DeiT-Ti and 0.31 for the other models. The blocks of even indexes except the $0^{th}$ block are selected to merge. By default, tokens' features ($\mathbf{X}_t$) are merged and then reordered to preserve the order.

### 3.2 COMPARATIVE EXPERIMENTS

**Comparative designs.** To validate both our theoretical claims and the practical effectiveness of R-MeeTo, we conduct a comparison of top-1 accuracy and FLOPs against two *state-of-the-art* token pruning techniques in Mamba: Token Recognition (Liang et al., 2022) and Hidden State Alignment (Zhan et al., 2024). Additionally, to assess the generality of R-MeeTo, we compare the performance of the re-trained VideoMamba (VideoM) (Li et al., 2024a) with token merged and the original pretrained one. Specifically, Vim-Ti and VideoM-Ti, due to their low capacity, are re-trained with token merging for 30 epochs. Contrarily, other models undergo re-training for 15 epochs. Additionally, following Vim (Zhu et al., 2024), Vim-B is re-trained using EMA with a 0.996 decay rate. **Analyses.** The comparison results are presented in Tab. 2. We observe that R-MeeTo

| method | top-1 acc.(%) | | | FLOPs (G) | | |
|---|---|---|---|---|---|---|
| | Vim-Ti | Vim-S | Vim-B | Vim-Ti | Vim-S | Vim-B |
| Vim (baseline) (Zhu et al., 2024) | 76.1 0.0 | 80.5 0.0 | 81.9 0.0 | 1.45 0.00 | 5.08 0.00 | 18.87 0.00 |
| Token Recognition (Liang et al., 2022) | 71.3 4.8↓ | 74.8 5.7↓ | - - | 1.28 0.17↓ | 3.57 1.51↓ | - - |
| Hidden State Alignment (Zhan et al., 2024) | 75.1 1.0↓ | 78.8 1.7↓ | - - | 1.29 0.16↓ | 3.60 1.48↓ | - |
| R-MeeTo (ours) | **75.3 0.8↓** | **79.9 0.6↓** | **81.3 0.6↓** | 1.28 0.17↓ | 3.58 1.50↓ | 13.21 5.66↓ |
| | PlainM-L1 | PlainM-L2 | PlainM-L3 | PlainM-L1 | PlainM-L2 | PlainM-L3 |
| PlainM (baseline) (Yang et al., 2024b) | 77.9 0.0 | 81.6 0.0 | 82.3 0.00 | 3.00 0.00 | 8.10 0.00 | 14.4 0.0 |
| Token Recognition (Liang et al., 2022) | 75.0 2.9↓ | 78.3 2.7↓ | 78.9 3.4↓ | 2.44 0.56↓ | 6.22 1.88↓ | 8.35 6.05↓ |
| Hidden State Alignment (Zhan et al., 2024) | **77.4 0.5↓** | 81.0 0.6↓ | 81.7 0.6↓ | 2.46 0.54↓ | 6.27 1.83↓ | 8.44 5.96↓ |
| R-MeeTo (ours) | 77.3 0.6↓ | **81.4 0.2↓** | **82.1 0.2↓** | 2.46 0.54↓ | 6.29 1.81↓ | 8.46 5.94↓ |
| | VideoM-Ti | VideoM-S | VideoM-B | VideoM-Ti | VideoM-S | VideoM-B |
| VideoM (baseline) (Li et al., 2024a) | 76.9 0.0 | 81.2 0.0 | 82.7 0.0 | 1.45 0.0 | 5.08 0.00 | 18.87 0.00 |
| R-MeeTo (ours) | 75.9 1.0↓ | 80.1 1.1↓ | 81.9 0.8↓ | 1.28 0.17↓ | 3.58 1.50↓ | 13.21 5.66↓ |

Table 2: Comparison between different token reduction methods on the performance of Vim-Ti, Vim-S and Vim-B in ImageNet-1K classification. R-MeeTo (ours) consistently achieves higher top-1 accuracy (%) than competing methods across various scales of Vims and VideoMs while maintaining comparable FLOPs. PlainMamba (PlainM) are from (Yang et al., 2024a).

consistently achieves higher top-1 accuracy than competing methods across various scales of Vim models while maintaining comparable FLOPs. Specifically, R-MeeTo demonstrates a substantial improvement over Token Recognition for Vim-Ti, achieving a 3.7% higher top-1 accuracy. For Vim-S, R-MeeTo outperforms both Token Recognition and Hidden State Alignment, with considerable gains, respectively. Moreover, R-MeeTo yields a notable reduction in FLOPs for Vim-B and VideoM-B, decreasing from 18.87G to 13.21G, with only a 0.6%/0.8% decrease in top-1 accuracy, respectively. Additionally, unlike competing methods that show decreased performance with larger models after token reduction, R-MeeTo effectively recovers performance across models of varying scales, highlighting its robustness.

## 3.3 FASTER MAMBA IN MINUTES

We conduct re-training experiments for 3 epochs on Vim-Ti, Vim-S, and Vim-B for ImageNet-1K classification. The experiments are performed on a single machine equipped with 8 NVIDIA

| hardware | Vim-Ti | Vim-S | Vim-B |
|---|---|---|---|
| $1 \times 8 \times$ H100 (single machine) | 16.2 | 25.2 | 57.6 |
| $2 \times 8 \times$ H100 (InfiniBand (Pfister, 2001)) | 8.1 | 12.9 | 30.6 |
| $4 \times 8 \times$ H100 (InfiniBand (Pfister, 2001)) | 4.2 | 6.8 | 16.9 |

Table 3: Wall time (**minutes**) of re-training Vim-Ti, Vim-S and Vim-B for 3 epochs on 3 hardwares. Give us minutes, we back a faster Mamba. Fig. 4 shows how fast it is.

H100 GPUs. Additionally, we conduct the same experiments on two and four machines, each with 8 NVIDIA H100 GPUs, connected via InfiniBand (Pfister, 2001). This setup allows us to evaluate the scalability and performance of R-MeeTo across different hardware configurations. Specifically, gradient accumulation is performed over two steps, with the per-GPU batch sizes set as follows: Vim-Ti at $2304 = 1152 \times 2$, Vim-S at $1408 = 704 \times 2$, and Vim-B at $512 = 256 \times 2$. This ensures optimal utilization of GPU memory for each model variant. We report the wall time (in minutes) for each re-training in Tab. 3. As shown, all Vims are re-trained within 60 minutes. Re-training the Vim-S model on $4 \times 8 \times$ H100 costs $\leq 10$ minutes only. Give us minutes, we back a faster Mamba.

## 3.4 ABLATION STUDY

**Case study on token order: odd-even shuffle.**

We first conduct a case study on tokens' order after merging. Specifically, we re-train Vim and DeiT models on the ImageNet-1K for 3 epochs using R-MeeTo, comparing results on whether tokens shuffle. The shuffle strategy is odd-even shuffle, *e.g.*, indexes from 0 to 3: $[0, 1, 2, 3] \rightarrow \{[0, 2], [1, 3]\} \rightarrow [0, 2, 1, 3]$. It works in Transformers (Bolya et al., 2023).

| \model | tiny | | | small | | | base | | |
|---|---|---|---|---|---|---|---|---|---|
| order | ✗ | ✓ | Δ | ✗ | ✓ | Δ | ✗ | ✓ | Δ |
| DeiT | 69.7 | 69.7 | 0.0↓ | 79.1 | 79.0 | 0.1↓ | 80.7 | 80.7 | 0.0↓ |
| Vim | 64.8 | 74.0 | 9.4↑ | 72.8 | 79.3 | 6.6↑ | 72.5 | 80.2 | 7.7↑ |

Table 4: Ablation study on the impact of token order's to top-1 accuracy (%) of DeiT and Vim using R-MeeTo. Δ represents the difference in performance between with and without reordering. Token reordering has minimal impact on the performance of DeiT models but plays a critical roles in the performance of Vim models.

**Analyses.** The final top-1 accuracy comparison is presented in Tab. 4. As shown, token re-ordering has minimal impact on the performance of DeiT models during re-training and token merging, with

no significant difference in top-1 accuracy when re-ordering is applied or omitted. In contrast, maintaining token order substantially enhances performance for Vim models, with comparable improvements for Vims, respectively. These findings validate our theoretical claims and highlight the critical role of maintaining token order in Vims, particularly for the smallest variant (*i.e.*, Vim-Ti), underscoring its importance in token reduction.

**Quantitative study on token order: shuffle ratio.** To further investigate the role of token order in Vims during token reduction, we first conduct experiments where only $r_s\%$ of the tokens' features ($\mathbf{X}_t$) are shuffled before each token reduction operation. Here, $r_s$ denotes the shuffle ratio, defined as the proportion of unordered features relative to the total number of features. Next, to explore how different level of shuffle ratios influence varying token reduction methods, we evaluate the performances of re-trained Vim-S models using both token pruning and merging. All models are re-trained for 3 epochs for consistency in comparison.

**Analyses.** Tab 6a shows the performance of training-free and re-trained Vims under different shuffle ratios. As observed, the performance of training-free Vims drastically declines as the randomness in the ordering of feature sequences increases. This emphasizes the importance of token order in the Vim architectures. This effect is particularly pronounced for Vim-Ti, likely due to its fewer number of parameters and lower capacity. Nonetheless, re-training substantially recovers performance for both Vim-Ti and Vim-S, validating the effectiveness of our proposed method (*i.e.*, R-MeeTo). On the other hand, the comparison of different token reduction operations across different shuffle ratios is illustrated in Tab 6b. It can be seen that token merging consistently outperforms token pruning for both training-free and re-trained models at all shuffle ratios. These results further support our theoretical conclusions and validate that better do token merging instead of pruning.

| model | feature | training-free | re-trained | $\Delta$ |
|---|---|---|---|---|
| Vim-Ti | $\mathbf{X}_t$ | 38.2 | 74.0 | 35.8↑ |
| | $\mathbf{C}_t$ | 34.8 | 73.8 | 39.0↑ |
| | $\mathbf{B}_t$ | 49.7 | 73.8 | 24.1↑ |
| | $\boldsymbol{\Delta}_t$ | 31.2 | 73.9 | 42.7↑ |
| Vim-S | $\mathbf{X}_t$ | 76.3 | 79.3 | 3.0↑ |
| | $\mathbf{C}_t$ | 76.1 | 78.7 | 2.6↑ |
| | $\mathbf{B}_t$ | 74.9 | 78.1 | 3.2↑ |
| | $\boldsymbol{\Delta}_t$ | 76.2 | 78.6 | 2.4↑ |

Table 5: Ablation study on the impact of different feature choices to top-1 accuracy (%) during token merging in R-MeeTo. $\Delta$ represents the difference in performance between training-free and re-trained models using selected feature. Token features ($\mathbf{X}_t$) can accurately summarizes the information within tokens in the Vim architecture.

| model | shuffle ratio | training-free | re-trained | $\Delta$ | | operation | shuffle ratio | training-free | re-trained | $\Delta$ |
|---|---|---|---|---|---|---|---|---|---|---|
| Vim-Ti | 0.1 | 24.0 | 69.9 | 45.9↑ | | pruning | 0.1 | 61.0 | 76.0 | 15.0↑ |
| | 0.3 | 8.3 | 65.5 | 57.2↑ | | | 0.3 | 33.0 | 72.6 | 39.6↑ |
| | 0.5 | 5.7 | 64.3 | 58.6↑ | | | 0.5 | 27.2 | 71.5 | 44.3↑ |
| | 0.7 | 5.1 | 63.9 | 58.8↑ | | | 0.7 | 25.6 | 71.4 | 45.8↑ |
| Vim-S | 0.1 | 61.3 | 76.2 | 14.9↑ | | merging | 0.1 | 61.3 | 76.2 | 14.9↑ |
| | 0.3 | 33.2 | 73.0 | 39.8↑ | | | 0.3 | 33.2 | 73.0 | 39.8↑ |
| | 0.5 | 27.3 | 71.9 | 44.6↑ | | | 0.5 | 27.3 | 71.9 | 44.6↑ |
| | 0.7 | 26.1 | 71.6 | 45.5↑ | | | 0.7 | 26.1 | 71.6 | 45.5↑ |

(a) **Varying shuffle ratio.** Token order is crucial.  (b) **Merging wins pruning** at all shuffle ratios.

Table 6: Ablation study on the impact of token order's on the performance of Vim-Ti and Vim-S in ImageNet-1K classification with varying shuffle ratios. Shuffle ratio represents the proportion of unordered token features relative to the total number of token features. $\Delta$ represents the difference in performance between training-free and re-trained models. Maintaining token order is important in Vim architecture for all token reduction methods, re-training can effectively recover the model performance after token reduction.

**Metric ablation: features.** To assess the impact of different features on measuring token similarity, we apply the token merging operation each to tokens' features ($\mathbf{X}_t$), the output features ($\mathbf{C}_t :=$ $C_t(\mathbf{X}_t)$), the input features ($\mathbf{B}_t := B_t(\mathbf{X}_t)$) and gated features ($\boldsymbol{\Delta}_t := \Delta_t(\mathbf{X}_t)$). Then, we re-train Vims on ImageNet-1K with each feature for 3 epochs.

**Analyses.** A previous study (Bolya et al., 2023) on token merging in ViT concluded that tokens' features ($\mathbf{X}_t$) are less effective than other features (*e.g.*, attention query, attention key, etc.) for determining token importance. However, within the Vim architecture, our findings reveal the opposite ones, as shown in Tab. 5. Specifically, Vim-Ti's performance after token reduction and re-training has little difference for all features, showing its compatibility.

Meanwhile, in Vim-S, employing $\mathbf{X}_t$ results in notably better performance, outperforming other features by up to 0.9%. These overall results suggest that $\mathbf{X}_t$ can accurately summarize the information within tokens in the architectures.

**Metric ablation: distance function.** To investigate the impact of different distance functions, we conduct experiments as follows. Detailedly, we select cosine similarity, $\ell_1$ distance, and $\ell_2$ distance as our distance functions for measuring token similarity in the token merging operation. we compare the final top-1 accuracies of re-trained Vims on ImageNet-1K using selected functions for 3 epochs.

| model | distance | training-free | re-trained | $\Delta$ |
|---|---|---|---|---|
| Vim-Ti | cosine | 38.2 | 74.2 | 36.0↑ |
| | $\ell_1$ | 38.1 | 74.2 | 36.1↑ |
| | $\ell_2$ | 38.2 | 74.3 | 36.1↑ |
| Vim-S | cosine | 76.3 | 79.3 | 3.0↑ |
| | $\ell_1$ | 76.3 | 79.3 | 3.0↑ |
| | $\ell_2$ | 76.3 | 79.4 | 3.1↑ |

Table 7: Ablation study on the impact of different distance function to top-1 accuracy (%) during token merging in R-MeeTo. $\Delta$ represents the difference in performance between training-free and re-trained models using selected distance function. The Vims show robustness to the choice of distance function in the token merging operation.

**Analyses.** The results are illustrated in Tab. 7. As observed, all three distance functions yield comparable top-1 accuracies, with only marginal differences across trials. Namely, no single distance function consistently outperforms the others across all experiments for both Vim-Ti and Vim-S models. These findings suggest that the Vim architectures demonstrate robustness to the choice of distance functions in the token merging operation, indicating flexibility in similarity metrics keeping performance, supporting our design.

**Re-training efficiency.** To evaluate the adaptability and scalability of our approach, we conduct ablation experiments by re-training Vims on the ImageNet-1K's smaller subsets. These subsets have 1%, 5%, and 10% data of the original ImageNet-1K dataset. The re-training phase sustains 3/subset ratio epochs, maintaining a consistent number of model update steps. Subsequently, to evaluate R-MeeTo's scalability over extended re-training, we re-train models on full ImageNet-1K varying #epochs.

| dataset | 100% | 10% | 5% | 1% |
|---|---|---|---|---|
| acc. (w/o re-train: 76.3) | 79.2 | 78.4 | 78.4 | 76.0 |
| $\Delta$ (Comparing to 76.3) | 2.9↑ | 2.1↑ | 2.1↑ | 0.3↓ |

Table 8: Ablation study on the impact of different dataset scales to top-1 accuracy (%) of Vim-S using R-MeeTo. $\Delta$ represents the difference in performance between training-free and re-trained models using a subset of the full ImageNet-1K dataset. R-MeeTo demonstrates adaptability to smaller datasets but becomes susceptible to over-fitting when the dataset is too limited in size.

**Analyses.** The results validating the adaptability and scalability of R-MeeTo are presented in Tab. 8 and Tab. 9. As shown in Tab. 8, R-MeeTo demonstrates the highest effectiveness on the full ImageNet-1K dataset, outperforming the performance of training-free model by 2.9%. Additionally, re-training on 5% subsets still achieves a 2.6% improvement over the training-free method, supporting R-MeeTo's adaptability on smaller datasets. However, re-training on only 1% subsets shows decreased performance compared to the training-free approach due to increased susceptibility to over-fitting. In contrast, longer re-training phases consistently enhance model performance after token merging, as seen in Tab. 9, confirming R-MeeTo's scalability with extended re-

| epoch | 1 | 3 | 5 | 7 | 15 |
|---|---|---|---|---|---|
| acc. ( w/o re-train: 76.3) | 79.0 | 79.3 | 79.5 | 79.6 | 80.0 |
| marginal benefit | 2.6 | 1.0 | 0.6 | 0.5 | 0.3 |
| minutes | 36 | 107 | 180 | 255 | 536 |

Table 9: Ablation study on the impact of re-training duration to top-1 accuracy (%) of Vim-S in ImageNet-1K classification using R-MeeTo. Marginal benefit represents the difference in performance between training-free and re-trained models for each additional epoch. Re-training for 1-5 epochs is the most cost-effective. Wall time reported is trained on 4 A100.

training epochs. Nevertheless, the incremental performance gains diminish with prolonged training, indicating that the benefits of extended re-training gradually reach a plateau. In summary, re-training for only 1-5 epochs provides the best trade-off between performance and computational cost.

**Inference throughput.** We comprehensively measure the empirical throughput (inference per second in float16 precision) and top-1 accuracy of Vim-S using R-MeeTo across various reduction ratios on NVIDIA RTX 3090, RTX 4090, V100, A4000, A100, and H100 GPUs to evaluate the efficiency and scalability of our approach on different hardware architectures. This benchmarking allows us to assess how well R-MeeTo adapts to both consumer-level and enterprise-level GPUs, offering opportunities into performance gains achieved by varying the reduction ratio.

**Analyses.** The results are detailed in Fig. 4. Notably, a slight decrease in top-1 accuracy occurs at a reduction ratio of 0.14, likely due to I/O and additional computational overhead surpassing the

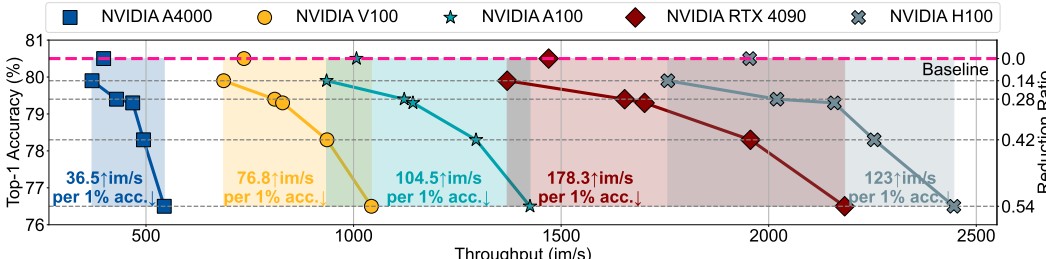

Figure 4: Throughput and top-1 accuracy comparison of Vim-S using R-MeeTo across different reduction ratios and GPUs. R-MeeTo effectively optimizes inference speed while preserving strong model accuracy across various hardware platforms. Notably, the performance drop at 0.14 ratio comes from I/O and additional computational overhead outweighing the benefits of token reduction.

token reduction benefits. For other reduction ratios, throughput is significantly enhanced across all tested GPUs, with only a marginal decrease in performance. These results validate the efficiency and scalability of our proposed method, demonstrating that R-MeeTo effectively optimizes inference speed. Meanwhile, it also maintains robust model accuracy across a range of hardware architectures, making it adaptable for both consumer-level, enterprise-level and other high-performance devices.

## 4 RELATED WORK

**Vision Mamba.** Mamba (Gu & Dao, 2023), based on SSM (Gu et al., 2022; Mehta et al., 2023; Fu et al., 2023; Smith et al., 2023), achieves a competitive performance with Transformer (Vaswani et al., 2017) with only linear complexity of #tokens. Recently, many works(Huang et al., 2024; Li et al., 2024a; Liu et al., 2024b; Patro & Agneeswaran, 2024; Pei et al., 2024; Yang et al., 2024a; Guo et al., 2024; Hatamizadeh & Kautz, 2024; Chen et al., 2024b; Shi et al., 2024; Ruan & Xiang, 2024) explore the effectiveness of Mamba in computer vision. The architectures of Mamba in computer vision have two main branches: 1) Vim series (Zhu et al., 2024; Huang et al., 2024; Li et al., 2024a), design a more effective bi-directional scanning block. 2) Vmamba (Liu et al., 2024b) series (Liu et al., 2024b; Yang et al., 2024a; Pei et al., 2024; Patro & Agneeswaran, 2024), focus on cross-scan. Moreover, different types of data are employed, *e.g.*, video (Li et al., 2024a;b; Yang et al., 2024c; Hu et al., 2024; Chen et al., 2024a), 3D (Zhang et al., 2024; Liang et al., 2024; Han et al., 2024; Liu et al., 2024a), multimodel (Shi et al., 2024; Li et al., 2024b; Dong et al., 2024; Qiao et al., 2024), motion sequence (Zhang et al., 2025). However, these works primarily focus on mechanisms and data, lacking further optimization of existing popular vision Mambas and theoretical supports.

**Token Reduction.** Token pruning, as a popular strategy to reduce tokens, has already demonstrated great potential in accelerating Transformers in both natural language processing (Goyal et al., 2020; Kim et al., 2022; Kim & Cho, 2021) and computer vision (Meng et al., 2022; Rao et al., 2021; Yin et al., 2022; Fayyaz et al., 2022; Song et al., 2022). However, directly deleting tokens inevitably loses the information of pruned tokens. Merging (Xu et al., 2022; Liang et al., 2022; Kong et al., 2022; Ryoo et al., 2021; Bolya et al., 2023; Marin et al., 2023; Chen et al., 2023), as an alternative of tokens reduction, preserves the information from discarded tokens. As an example of merging in Transformers, ToMe (Bolya et al., 2023) achieves inference acceleration without training. However, Mamba has fundamental differences from Transformers, making it challenging to apply methods from Transformers to Mamba. The difference comes from the sequence dependency of tokens in SSM. The most related work is Hidden State Alignment (Zhan et al., 2024), which designs a selective skipping mechanism to choose pruned tokens in Vims. This work focuses on pruning methods in Mamba. Besides Hidden State Alignment (Zhan et al., 2024), a limited number of works currently reveal the essential cause of performance dropping by token reduction. The method about merging and its usage in Mamba is even less. Our work provides analyses about the main causes, availability of merging, and further gives effective solutions for both accelerating and recovering pruned Mambas' performance.

## 5 CONCLUSION

In this paper, we propose three main observation about token reduction in Mamba: 1) Mamba is more sensitive in token reduction; 2) Merging keep more key knowledge than pruning; 3) re-training can recover the dropped performance simply and efficiently. From the perspective of information transfer, we analyses the main causes of the sensitivity and performance drop are the *enrichment effect*, the imbalance knowledge storage in Mamba, and the *loss of key knowledge*. Empirically, we provide verification about our theory and the compatibility and efficiency of the proposed recovering strategy, *i.e.* R-MeeTo, on token reduction.

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

## A    Detailed Experiment Settings

**Token reduction ratio.** We use a reduction number of each layer $r$ to control the whole token reduction ratio. Table 10 shows the relationship between the reduction of each layer and the token reduction ratio.

| $r$ | 0 | 5 | 10 | 11 | 13 | 15 | 20 |
|---|---|---|---|---|---|---|---|
| reduction ratio | 0 | 0.14 | 0.28 | 0.31 | 0.36 | 0.42 | 0.54 |

Table 10: Relationship between reduction number of tokens $r$ for each layer and the reduction ratio.

In our experiments, our token reduction strategy is to reduce $r$ tokens each time, and the ablation of $r$, *w.r.t.* performance, is shown in Table 11.

| model | $r$ the number of reduced tokens per layer | | | | | | | | | | | |
|---|---|---|---|---|---|---|---|---|---|---|---|---|
| | 0 | 1 | 2 | 3 | 4 | 5 | 6 | 7 | 8 | 9 | 10 | 11 |
| Vim-Ti | 76.1 | 44.9 | 52.3 | 40.3 | 44.5 | 38.2 | 41.7 | 37.0 | 40.0 | 35.6 | 38.4 | 35.5 |
| Vim-S | 80.5 | 77.4 | 78.0 | 78.7 | 78.4 | 78.5 | 78.3 | 78.1 | 77.7 | 77.5 | 76.7 | 76.3 |
| Vim-B | 80.3 | 79.1 | 78.6 | 78.3 | 78.8 | 78.4 | 78.7 | 78.5 | 78.6 | 78.3 | 78.4 | 78.3 |

Table 11: Ablation of pruning hyperparameter $r$: deciding reduction ratio↑. Larger $r$ means larger reduction ratio. Top-1 accuracy (%) is reported.

**Token-reduced blocks.** In Fig. 1, the Mamba and Transformers are all pruned and merged by the same reduction ratio. The $r$ number of pruned and merged blocks is layer-wise. Other experiments without comparison with Transformer in the main paper is empirically using even-block reduction, where the first block (indexed as 0) is kept and the other even-indexed blocks are token-reduced. The intervals between reduced tokens are 2 blocks as default, which means that we reduce tokens every 2 blocks. *Additional experiments* using odd-block reduction are also included in Section. D.3 of Appendix, where the odd-indexed blocks are token-reduced.

## B    Implementation Details

The framework of R-MeeTo is shown in Algorithm 1, and the detailed modules are distributed in Algorithm 2, Algorithm 3 and Algorithm 4.

Our R-MeeTo is consists of 2 main modules: token reduction and re-training. The first module follows ToMe (Bolya et al., 2023), but our implementation is on Mamba instead of Transformer. We therefore propose our token merge method in the algorithms and simply use re-training to recover performance. The intuitions of each process are presented in the comments.

## C    Analyses Details

### C.1    Explanation

**Tokens order.** The token order means that before merge, we save the time $t$ order of the tokens, and after merge, we sort the tokens in the original time $t$ order (default setting).

**Features in Tab. 5.** The feature is obtained by the outputs of components in Mamba, including outputs of bi-directional SSM and linear projection $\mathbf{X}_t$, output features of bi-directional SSM $\mathbf{C}_t$, hidden states' increments $\mathbf{B}_t$ and gated features $\mathbf{\Delta}_t$.

**Less data and less training iteration.** Re-training on subset (with less data). We re-train Vim-S on 1%, 5%, 10% subset of ImageNet-1K. We maintain the same iterations for all subsets. Re-training with less iterations. We re-train Vim-S with 1/3/5/7 epochs on full ImageNet-1K.

---

**Algorithm 1** Re-training Merged Tokens (R-MeeTo).

---

1: **Input:** model: $M_\theta$, dataset: $D$, distance: dist.$(\cdot, \cdot)$.
2: **Output:** a faster model.
3: ▯▯▯▯▯   $\text{tkns}^{(l)} \leftarrow \{\text{tkn}_t^{(l)}\}_{t \in [N_{\text{total}}]} := M_\theta^{(l)}(D), l \in [L]$;
4:                                                        ▷ *M has L layer to reduce.*
5:                                      ▷ *l is hidden below, ∵ we do the same for each layer.*
6: **–   Token Reduction   –**
7: ▯▯▯ ▯▯▯   $\text{tkns}_1, \text{tkns}_2 \leftarrow \text{Grouping}(\text{tkns})$;
8:                                   ▷ *tkns,tkns₁,tkns₂: the input, 1-st and 2-nd divided tokens*
9: ▯ $\xleftrightarrow{\text{dist.}}$ ▯   $\text{dists} \leftarrow \{\text{dist}_{i,j} := \text{dist.}(\text{tkn}_i, \text{tkn}_j)\}$,
10:                $\forall\, \text{tkn}_i, \text{tkn}_j \in \text{tkns}_1, \text{tkns}_2$;
11:                                            ▷ *Calculate distance between tokens in the groups.*
12: ▯▯▯ ▮▯   $\text{toMs} \leftarrow \{(i,j)| -\text{dist}_{i,j} \in \text{top-}r(-\text{dists})\}$;
13:                                              ▷ *Take the r closest: tokens to merge.*
14: ▮▯▯   $\hat{\text{tkns}}^{(l+1)} \leftarrow \{\hat{\text{tkn}}_t^{(l)}\}_{t \in [N_{\text{total}} - r \times l]}$
15:      $:= \text{Merge}(\text{toMs})$;                                  ▷ *t̂kn: the next layer's inputs;*
16: **–      Re-training      –**
17: $\hat{\theta} \leftarrow$ Use dataset $D$ to re-train $\theta$ for given epochs.
18: **Return** reduced and re-trained $\hat{\theta}$

---

**Algorithm 2** Our Implementation: Grouping.

---

1: **Input:** given tokens: tkns.
2: **Output:** grouped tokens: $\text{tkns}_1, \text{tkns}_2$.
3: ▯▯▯▯   $\text{tkns}_1 \leftarrow \{\text{tkn}_i | i\%2 = 0, i \in [\text{idx}(\text{tkns})]\}$
4: ▯▯▯▯   $\text{tkns}_2 \leftarrow \{\text{tkn}_i | i\%2 = 1, i \in [\text{idx}(\text{tkns})]\}$
5:                                       ▷ *Divide tokens into odd and even indexes.*
6: **Return** $\text{tkns}_1, \text{tkns}_2$

---

**Shuffle ratio.** *Shuffle ratio* is defined as follows:

$$\text{shuffle ratio} = N_{\text{shuffled}}/N_{\text{total}}, \tag{8}$$

where $N_{\text{shuffled}}$ refers to the number of shuffled tokens. $N_{\text{total}}$ refers to the total number of tokens. Higher shuffle ratio means more serious disruption to tokens' order.

## C.2   THEORETICAL ANALYSES

**Proposition 2.** *(No dependency before in SSM's tokens.) In a SSM block, as a sequential components, its outputs $Y_{0:t}$ before time $t$ do not have dependency on the inputs $X_t$ at $t$. Thus, we have:*

$$I(Y_{0:t}^{\mathbf{M}}; X_t) = 0, \forall t \in [T].$$

**Definition 1.** *(Mutual information) Mutual information between 2 signals is defined with Shannon entropy $H$ as:*

$$I(X;Y) = H(X) - H(X|Y)$$
$$= I(Y;X) = H(Y) - H(Y|X)$$

**Definition 2.** *(Interaction information's definition.) Interaction information between 3 signals is defined with Shannon entropy $H$ as:*

$$I(X;Y;Z) := H(X) + H(Y) + H(Z) + H(X,Y,Z)$$
$$- H(X,Y) - H(X,Z) - H(Y,Z).$$

**Lemma 1.** *(Equal total knowledge.) The amount of total knowledge, compressed by Attention and SSM blocks from inputs $X_t$ at time $t$, is the same if Assumption 1 holds. $I(Y_{0:T}^{\mathbf{T}}; X_t) = I(Y_{0:T}^{\mathbf{M}}; X_t), \forall t \in [T]$*

---

**Algorithm 3** Our Implementation: Distance.

---

1: **Input:** given token groups: $\text{tkns}_1, \text{tkns}_2$.
2: **Output:** distance matrix between tokens: dists .
3: $\| \overset{1-\cos}{\longleftrightarrow} \| \{\text{dist}_{i,j} := 1 - \cos(\text{tkns}_1, \text{tkns}_2)\}$,
4: $\qquad \forall\, \text{tkn}_i \in \text{tkns}_1, \text{tkn}_j \in \text{tkns}_2$
5: $\qquad\qquad\qquad\qquad\qquad\qquad$ ▷ *Calculate cosine distance between tokens:* $1 - \cos$
6: **Return** $\{\text{dist}_{i,j}\}$

---

**Algorithm 4** Out Implementation: Merge.

---

1: **Input:** tokens to merge: toMs, source tokens: tkns
2: **Output:** merged tokens: $\{\hat{\text{tkn}}_t\}$
3: $\hat{\text{tkns}} \leftarrow \{\text{tkn}_i + \text{tkn}_j | (i,j) \in \text{toMs}\}$
4: $\qquad\qquad\qquad\qquad\qquad\qquad\qquad\qquad$ ▷ *Add tokens together.*
5: $\hat{\text{tkns}} \leftarrow \hat{\text{tkns}} \cup \{\text{tkn}_i \in \text{tkns} | (\cdot, i) \wedge (i, \cdot) \notin \text{toMs}\}$
6: $\qquad\qquad\qquad\qquad\qquad\qquad\qquad\qquad$ ▷ *Update token set.*
7: **Return** $\|\cup\|\,\| \hat{\text{tkns}}$ $\qquad\qquad\qquad\qquad\qquad\qquad$ ▷ *Keep Order.*

---

*Proof.*

$$
\begin{aligned}
I(Y_{0:T}^{\mathbf{T}}; X_t) &= H(Y_{0:T}^{\mathbf{T}}) - H(Y_{0:T}^{\mathbf{T}}|X_t) \\
&= H(Y_{0:T}^{\mathbf{M}}) - H(Y_{0:T}^{\mathbf{M}}|X_t) \\
&= I(Y_{0:T}^{\mathbf{M}}; X_t).
\end{aligned}
$$

$\qquad\qquad\qquad\qquad\qquad\qquad\qquad\qquad\qquad\qquad\qquad\qquad\qquad\qquad\qquad$ □

Theorem 1 is proven as followings:

*Proof.* Since the second inequality is simply introduced by the first, we mainly prove the inequality for the first, *i.e.*:

$$
I(Y_{t:T}^{\mathbf{M}}; X_t) \geq I(Y_{t:T}^{\mathbf{T}}; X_t), \forall t \in [T]
$$

We add the both sides of the inequality as follows:

$$
\begin{aligned}
&I(Y_{t:T}^{\mathbf{T}}; X_t) + I(Y_{0:t}^{\mathbf{T}}; X_t) \\
&\qquad = \underbrace{I(Y_{0:T}^{\mathbf{T}}; X_t)}_{\text{total knowledge}} + \underbrace{I(Y_{0:t}^{\mathbf{T}}; Y_{t:T}^{\mathbf{T}}; X_t)}_{\text{interaction info.}} \\
&I(Y_{t:T}^{\mathbf{M}}; X_t) + I(Y_{0:t}^{\mathbf{M}}; X_t) \\
&\qquad = \underbrace{I(Y_{0:T}^{\mathbf{M}}; X_t)}_{\text{total knowledge}} + \underbrace{I(Y_{0:t}^{\mathbf{M}}; Y_{t:T}^{\mathbf{M}}; X_t)}_{\text{interaction info}}.
\end{aligned}
$$

With Lemma 1, let $\mathcal{K}$ represents the total knowledge, we have:

$$
I(Y_{t:T}^{\mathbf{T}}; X_t) = I(Y_{0:T}^{\mathbf{T}}; X_t) + I(Y_{0:t}^{\mathbf{T}}; Y_{t:T}^{\mathbf{T}}; X_t) - \underbrace{I(Y_{0:t}^{\mathbf{T}}; X_t)}_{\geq 0} \tag{9}
$$

$$
I(Y_{t:T}^{\mathbf{M}}; X_t) = I(Y_{0:T}^{\mathbf{M}}; X_t) + I(Y_{0:t}^{\mathbf{M}}; Y_{t:T}^{\mathbf{M}}; X_t),
$$

where we only need to discuss the first two terms for the targeted inequality. The first term, called total knowledge, is discussed in Lemma 1, and the second term is the interaction information between $X_t$, $Y_{0:t}$ and $Y_{t:T}$.

$$
\begin{aligned}
&I(X_t, Y_{0:t}^{\mathbf{T}}, Y_{t:T}^{\mathbf{T}}) \\
&= H(X_t) + H(Y_{0:t}^{\mathbf{T}}) + H(Y_{t:T}^{\mathbf{T}}) + H(X_t, Y_{0:T}^{\mathbf{T}}) \\
&\quad - H(X_t, Y_{0:t}^{\mathbf{T}}) - H(X_t, Y_{t:T}^{\mathbf{T}}) - H(Y_{0:T}^{\mathbf{T}}, Y_{t:T}^{\mathbf{T}})
\end{aligned}
$$

Reorganize this equation, we have:

$$I(X_t; Y_{0:t}^{\mathbf{T}}; Y_{t:T}^{\mathbf{T}})$$

$$= \underbrace{H(X_t) + H(Y_{t:T}^{\mathbf{T}}) - H(X_t, Y_{t:T}^{\mathbf{T}})}_{(1)}$$

$$- \underbrace{[H(Y_{0:t}^{\mathbf{T}}, Y_{t:T}^{\mathbf{T}}) - H(Y_{0:t}^{\mathbf{T}})]}_{(2)}$$

$$+ \underbrace{H(X_t, Y_{0:T}^{\mathbf{T}}) - H(X_t, Y_{0:t}^{\mathbf{T}})}_{(3)},$$

where each term has a tranform as follows:

$$(1) = I(X_t; Y_{t:T}^{\mathbf{T}})$$
$$= H(X_t) - H(X_t | Y_{t:T}^{\mathbf{T}})$$
$$(2) = H(Y_{t:T}^{\mathbf{T}} | Y_{0:t}^{\mathbf{T}})$$
$$(3) = H(Y_{t:T}^{\mathbf{T}} | Y_{0:t}^{\mathbf{T}}, X_t).$$

We have:

$$I(X_t; Y_{0:t}^{\mathbf{T}}; Y_{t:T}^{\mathbf{T}}) = I(X_t; Y_{t:T}^{\mathbf{T}}) - (2) + (3)$$
$$= I(Y_{t:T}^{\mathbf{T}}; X_t) - (2) + (3)$$
$$= [H(Y_{t:T}^{\mathbf{T}}) - (2)] - [H(Y_{t:T}^{\mathbf{T}} | X_t) - (3)]$$

i) With Assumption 2, we simply have:

$$I(X_t; Y_{0:t}^{\mathbf{T}}; Y_{t:T}^{\mathbf{T}}) = I(Y_{0:t}^{\mathbf{T}}; Y_{t:T}^{\mathbf{T}}) - I(Y_{0:t}^{\mathbf{T}}; Y_{t:T}^{\mathbf{T}} | X_t).$$
$$= I(Y_{0:t}^{\mathbf{M}}; Y_{t:T}^{\mathbf{M}}) - I(Y_{0:t}^{\mathbf{M}}; Y_{t:T}^{\mathbf{M}} | X_t).$$
$$= I(X_t; Y_{0:t}^{\mathbf{M}}; Y_{t:T}^{\mathbf{M}})$$

Thus, according to Lemma 1 and Equ. 9, the inequality holds.

ii) However, with Proposition 2 and without Assumption 2, the followings hold:

$$\underbrace{I(X_t; Y_{t:T}^{\mathbf{T}} | Y_{0:t}^{\mathbf{T}}) + I(Y_{0:t}^{\mathbf{T}}; Y_{t:T}^{\mathbf{T}}; X_t)}_{I(X_t; Y_{t:T}^{\mathbf{T}})} + \underbrace{I(X_t; Y_{0:t}^{\mathbf{T}} | Y_{t:T}^{\mathbf{T}})}_{\geq 0}$$

$$= \underbrace{I(X_t; Y_{t:T}^{\mathbf{M}} | Y_{0:t}^{\mathbf{M}})}_{I(X_t; Y_{t:T}^{\mathbf{M}})} + \underbrace{I(Y_{0:t}^{\mathbf{T}}; Y_{t:T}^{\mathbf{T}}; X_t)}_{=0} + \underbrace{I(X_t; Y_{0:t}^{\mathbf{M}} | Y_{t:T}^{\mathbf{M}})}_{=0},$$

where the equation holds is because of non-dependent tokens and Lemma 1. Interaction information has term

$$(3) - (2) = -I(X_t; Y_{t:T}^{\mathbf{T}} | Y_{0:t}^{\mathbf{T}}),$$

and this means the key knowledge is transferred between tokens $Y_{0:T}^{\mathbf{T}}$, leading a trade-off and balance between storing knowledge in Transformer's tokens of $0 : t$ or $t : T$.

The inequality is thus proven, notice that the time $t$ is not specified in our proven, and the second equation can be obtained by the reductio ad absurdum. □

## D  MORE EXPERIMENTS

### D.1  EXPERIMENTS ON VIDEOS

**Comparative experiment on VideoMamba.** We perform experiments on the Kinetics-400 (K400) video dataset (Carreira & Zisserman, 2017) to further evaluate our R-MeeTo.

| reduction ratio | | RTX 3090 | | RTX 4090 | | V100 | | A4000 | | A100 | | H100 | |
|---|---|---|---|---|---|---|---|---|---|---|---|---|---|
| | acc. (%) | im/s | speed | im/s | speed | im/s | speed | im/s | speed | im/s | speed | im/s | speed |
| 0.00 | 80.5 | 739 | 1.00 × | 1470 | 1.00 × | 736 | 1.00 × | 398 | 1.00 × | 1007 | 1.00 × | 1954 | 1.00 × |
| 0.14 | 79.9 | 706 | 0.96 × | 1370 | 0.93 × | 687 | 0.93 × | 370 | 0.93 × | 935 | 0.93 × | 1757 | 0.90 × |
| 0.28 | 79.4 | 825 | 1.12 × | 1653 | 1.12 × | 810 | 1.10 × | 429 | 1.08 × | 1122 | 1.11 × | 2020 | 1.03 × |
| 0.31 | 79.3 | 892 | 1.21 × | 1701 | 1.16 × | 829 | 1.13 × | 468 | 1.18 × | 1143 | 1.14 × | 2158 | 1.10 × |
| 0.42 | 78.3 | 936 | 1.27 × | 1956 | 1.33 × | 936 | 1.27 × | 494 | 1.24 × | 1295 | 1.29 × | 2254 | 1.15 × |
| 0.54 | 76.5 | 1029 | 1.39 × | 2183 | 1.49 × | 1043 | 1.42 × | 544 | 1.37 × | 1425 | 1.42 × | 2446 | 1.25 × |

Table 12: Throughput and top-1 accuracy comparison of Vim-S using R-MeeTo across different reduction ratios and GPUs. R-MeeTo effectively optimizes inference speed while preserving strong model accuracy across various hardware platforms. Notably, the performance drop at the reduction ratio of 0.14 results from I/O and additional computational overhead outweighing the benefits of token reduction.

| method | top-1 acc.(%) | | | FLOPs (G) | | |
|---|---|---|---|---|---|---|
| | VideoM-Ti | VideoM-S | VideoM-M | VideoM-Ti | VideoM-S | VideoM-M |
| VideoM (Baseline) | 76.9  0.0 | 79.3  0.0 | 80.5  0.0 | 11.54  0.09 | 40.43  0.00 | 115.33  0.00 |
| R-MeeTo (training-free) | 75.4  1.5↓ | 77.5  1.8↓ | 78.4  2.1↓ | 9.49  2.05↓ | 30.99  12.33↓ | 71.71  43.62↓ |
| R-MeeTo (re-train) | 76.5  0.4↓ | 78.5  0.8↓ | 78.9  1.6↓ | 9.49  2.05↓ | 30.99  12.33↓ | 71.71  43.62↓ |

Table 13: Comparison between VideoMamba (Li et al., 2024a) with and without R-MeeTo on the performance of short-term understanding on Kinetics-400 (Carreira & Zisserman, 2017) classification. R-MeeTo yields a notable reduction in FLOPs while only leading to a slight accuracy drop. Top-1 accuracy (%) and GFLOPs are reported.

**Settings.** The clip is set to 8 frames per video. We set $r$=88 for VideoM-Ti/S/M. We apply the base learning rate of 2e-5 and re-training epochs of 30 for VideoM-Ti and 15 for VideoM-S/M. The top-1 accuracy and FLOPs are reported. We merge for every two blocks of the models. We merge 11 times in VideoM-Ti and VideoM-S, 15 times in VideoM-M.

**Analyses.** The reported comparison results are shown in Tab. 13. The R-MeeTo decreases top-1 accuracy slightly with notably reduced FLOPs. Specifically, R-MeeTo reduces commendable GFLOPs for Videom-Ti/S/B respectively, with only minimal performance drop. Note that the exact reduction ratio of VideoM-Ti/S is 0.31, while 0.42 for VideoM-M, because the depth of VideoM-Ti/S is 24 while the depth of VideoM-M is 32.

**Ablation study on $r$.** We conduct an ablation study on $r$ token merging number per layer in VideoM-Ti/S/M. We evaluate the models in training-free setting by the top-1 accuracy on Kinetics-400 (Carreira & Zisserman, 2017). The clip is 8 frames per video.

**Analyses.** The results are shown in Tab. 14. Our method seems more robust on video tasks than

| model / $r$ | 0 | 8 | 16 | 24 | 32 | 40 | 48 | 56 | 64 | 72 | 80 | 88 |
|---|---|---|---|---|---|---|---|---|---|---|---|---|
| VideoM-Ti | 76.89 | 76.94 | 76.95 | 76.92 | 76.95 | 76.80 | 76.61 | 76.48 | 76.23 | 75.98 | 75.81 | 75.45 |
| VideoM-S | 79.28 | 79.23 | 79.16 | 79.13 | 79.10 | 78.97 | 78.90 | 78.68 | 78.43 | 78.02 | 77.86 | 77.41 |
| VideoM-M | 80.47 | 79.78 | 79.68 | 79.74 | 79.70 | 79.51 | 79.39 | 79.31 | 79.12 | 78.90 | 78.75 | 78.44 |

Table 14: Ablation of pruning hyperparameter $r$: deciding reduction ratio↑. Larger $r$ means larger reduction ratio. Top-1 accuracy (%) is reported

on image tasks. The underlying reason can be the data redundancy in videos is larger than that in images.

## D.2 ABLATION STUDY ON MERGING

**General settings.** We set $r = 5$ for Vim-Ti and $r = 11$ for Vim-S. Learning rate decreases from 2e-5 to 1e-6 by cosine scheduler during re-training. The number of training epochs is 3. Top-1

| model | $n$ (the $n$-th closest to merge) | | | | |
|---|---|---|---|---|---|
| | 1-st | 3-rd | 5-th | 7-th | 14-th |
| Vim-Ti (Zhu et al., 2024) | 74.1 | 74.1 | 74.2 | 74.1 | 74.0 |
| Vim-S (Zhu et al., 2024) | 79.2 | 79.2 | 79.1 | 79.0 | 78.7 |

Table 15: Quantitative study on token similarity. The similar number means the order of matching pairs of one chosen-merged token. As we choose the less similar pairs to be merged, the performance on top-1 accuracy (%) of ImageNet-1K (Deng et al., 2009) decreases.

| model | interval $k$ (merge every $k$ layer) | | | |
|---|---|---|---|---|
| | 2 | 4 | 6 | 8 |
| Vim-Ti (Zhu et al., 2024) | 74.2 | 74.1 | 74.2 | 74.1 |
| Vim-S (Zhu et al., 2024) | 79.2 | 79.2 | 79.3 | 79.4 |

Table 16: Ablation study on merging step. We merge tokens for every *merging step*. We compare the performance on top-1 accuracy (%) of ImageNet-1K (Deng et al., 2009). The performance of R-MeeTo is maintained if the token ratio is the same.

accuracy (%) on ImageNet-1K (Deng et al., 2009) is reported. Block reduction number is 11 in Vim-Ti/S.

**Ablation on merging $n$-th closest token.** We conduct the experiment using the closet number of 1-st, 3-rd, 5-th, 7-th, and 14-th. We re-train the model for 3 epochs. The other settings are the same as the default settings.

**Analyses.** The results are shown in Tab. 15. The fact that the 1st-14th close tokens used for merging have little impact suggests that the similarities are actually very common in a wide range of tokens. There are actually a lot of similar tokens, and the redundancy is very large, supporting the overall intent of token reduction.

**Ablation on layer-wise intervals.** Here, we merge tokens in Vim-Ti/S by different intervals. We set the $r \in [5, 11, 18, 28]$ in Vim-Ti and $r \in [11, 24, 40, 62]$ in Vim-S to maintain the reduction ratio of Vim-Ti/S as 0.14/0.31. We fix #output tokens in Vim-Ti/S as 142/76.

**Analyses.** The reported results are all shown in Tab. 16. Larger intervals result in larger token reduction granularity. Larger granularity leads can introduce extra noise and decreases performance. Because of keeping the same number of reduced parameters, a larger interval make fewer reduction times, resulting in more tokens being reduced at once.

**Ablation on token merging operations.** To further explore an effective way for merging two token pairs. We conduct the ablation study on four merging operations: sum, mean, max pool, and min pool.

**Analyses.** The results are presented in Tab. 17. We observe that in both Vim-Ti and Vim-S, after re-training, different operation has limited impacts on performance.

D.3   ODD-BLOCK REDUCTION

**Comparison between token pruning and merging.** We conduct the training-free experiment on Vim-S. We report the top-1 accuracy on ImageNet-1K (Deng et al., 2009).

**Analyses.** The results are shown in Tab. 18. Merging consistently outperforms pruning on both even-block and odd-block reduction settings.

**Comparison between training-free and re-training.** We conduct the training-free and re-training experiment on Vim-S. We use a odd-block reduction merging operation.

**Analyses.** The results are shown in Tab. 19. We observe that re-training consistently enhances the performance on both even-block and odd-block reduction settings.

| model | merge op. | training-free | re-trained | $\Delta$ |
|---|---|---|---|---|
| | sum | 38.2 | 74.1 | 35.9↑ |
| Vim-Ti (Zhu et al., 2024) | mean | 41.8 | 74.1 | 32.3↑ |
| | max | 43.4 | 74.2 | 30.8↑ |
| | min | 39.7 | 74.2 | 34.5↑ |
| | sum | 76.3 | 79.2 | 2.9↑ |
| Vim-S (Zhu et al., 2024) | mean | 76.2 | 79.3 | 3.1↑ |
| | max | 74.7 | 79.5 | 4.8↑ |
| | min | 74.5 | 79.4 | 4.9↑ |

Table 17: Ablation study on the impact of different token merging methods on top-1 accuracy (%) in R-MeeTo. $\Delta$ represents the difference in performance between training-free and re-trained models. The max pool and min pool methods will performance better than the sum and mean merging methods after re-training.

| reduction ratio | pruning | merging | $\Delta$ |
|---|---|---|---|
| 0.14 | 78.3 | 78.4 | 0.1↑ |
| 0.28 | 76.1 | 76.3 | 0.2↑ |
| 0.42 | 69.9 | 71.5 | 1.6↑ |
| 0.54 | 44.9 | 54.6 | 9.7↑ |

Table 18: Comparison on the performance of Vim-S between token pruning and merging operations with odd-block reduction. Merging consistently outperforms pruning on both even-block and odd-block reduction settings. Top-1 accuracy (%) is reported.

**Comparative experiment.** We re-train Vim-Ti/S for 30/15 epochs respectively with R-MeeTo using odd-block reduction. We use a batchsize of 128 with gradient accumulation performed over two steps, and total batchsize of 1024= $4\times128\times2$. Models are trained with AdamW (Loshchilov & Hutter, 2019) optimizer with a learning rate decaying from 2e-5 to 1e-6 using a cosine scheduler, and a weight decay of 5e-2.

**Analyses.** The result is shown in Tab. 21. The difference between even-block and odd-block reduction operations is limited compare to the results in the main paper.

### D.4 ABLATE MODULES

**Settings.** We conduct a series of ablation studies to systematically evaluate the impact of key modules in Algorithm 1. Specifically, we modify the default operations in the algorithm to study the impacts on the overall performance:

1) We replace the default top-$r$ selection mechanism in Line 10 of Algorithm 1 with a random-$r$ approach. This change allows us to assess the importance of selecting the top-$r$ operations versus randomly choosing $r$ candidate tokens. 2) Next, we alter the default token pairs to merge in Line 3 of Algorithm 4 by replacing it with a random pair selection strategy. This modification helps us evaluate the effectiveness of the default merging strategy compared to a purely random pairing approach. 3) Finally, we ablate the impact of the Grouping $(\cdot)$ operation in Algorithm 1. By replacing this component, we aim to understand how the grouping mechanism contributes to the overall performance of the algorithm.

**Analyses.** The results are shown in Tab. 22 and Tab. 20 Dropping any module or introducing randomness lead new noise, which makes performance degrade. This proves the necessity of our individual modules.

| reduction ratio | training-free | re-trained | $\Delta$ |
|---|---|---|---|
| 0.14 | 78.4 | 80.0 | 1.6↑ |
| 0.28 | 76.3 | 79.2 | 2.9↑ |
| 0.42 | 71.5 | 78.0 | 6.5↑ |
| 0.54 | 54.6 | 75.6 | 21.0↑ |

Table 19: Comparison of the performance of Vim-S between the training-free and re-trained model with odd-block reduction. Re-training consistently enhances the performance on both even-block and odd-block reduction settings. Top-1 accuracy (%) is reported.

| model | Grouping(·) | training-free | re-trained | $\Delta$ |
|---|---|---|---|---|
| | odd-even | 38.2 | 74.1 | 35.9↑ |
| Vim-Ti (Zhu et al., 2024) | front-behind | 29.0 | 72.9 | 43.9↑ |
| | random | 42.1 | 73.0 | 30.9↑ |
| | odd-even | 76.3 | 79.2 | 2.9↑ |
| Vim-S (Zhu et al., 2024) | front-behind | 72.6 | 76.8 | 4.2↑ |
| | random | 72.6 | 77.1 | 4.5↑ |

Table 20: **Grouping Ablation**. Ablate Grouping(·) operation in Algorithm 2. Grouping is based on the indexes, time $t$. *Odd-even*: splitting the tokens into two groups according to their odd-even indexes. *Front-behind*: splitting the tokens into the first half part and the last half part. *Random*: randomly splitting all tokens into two groups.

## D.5    VISUALIZATION.

**Settings.** We provide visualization results on the ImageNet-1K dataset (Deng et al., 2009) and K-400 (Carreira & Zisserman, 2017) using a Vim-S and VideoM-S re-trained by R-MeeTorespectively, with $r = 10$ achieving top-1 accuracy of 79.9 and 78.5. These visualizations aim to demonstrate its effectiveness and provide qualitative insights into its decision-making process, supporting the quantitative results in the main paper.

**Analyses.** We observe that the image tokens belonging to the same object are successfully merged into a single group. This phenomenon suggests that R-MeeTo can accurately merge image tokens that exhibit similar features in reality. This capability not only validates the effectiveness of the method in feature extraction and matching but also demonstrates its robustness and adaptability in handling complex image scenes.

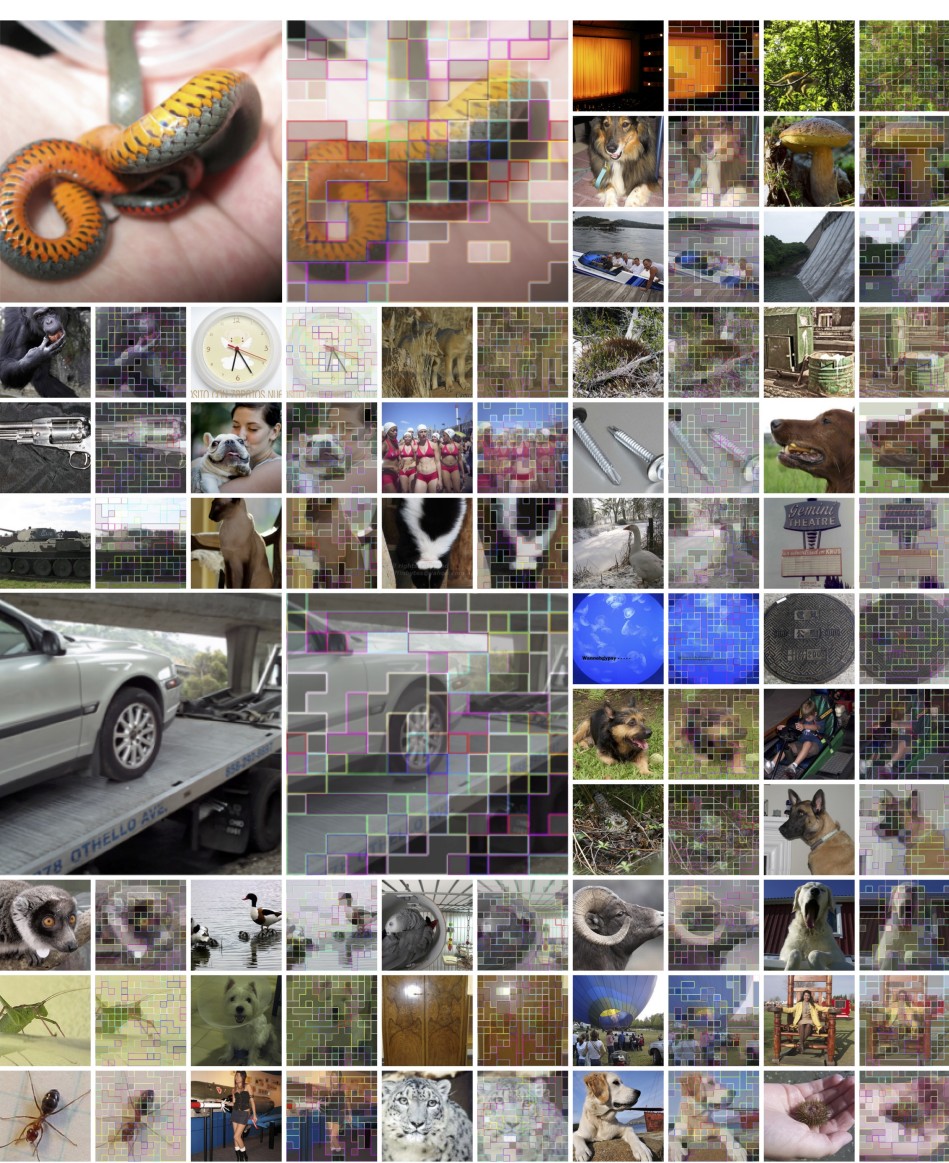

Figure 5: Visualization of R-MeeTo on ImageNet-1K (Deng et al., 2009). Tokens belonging to one object are merged into one.

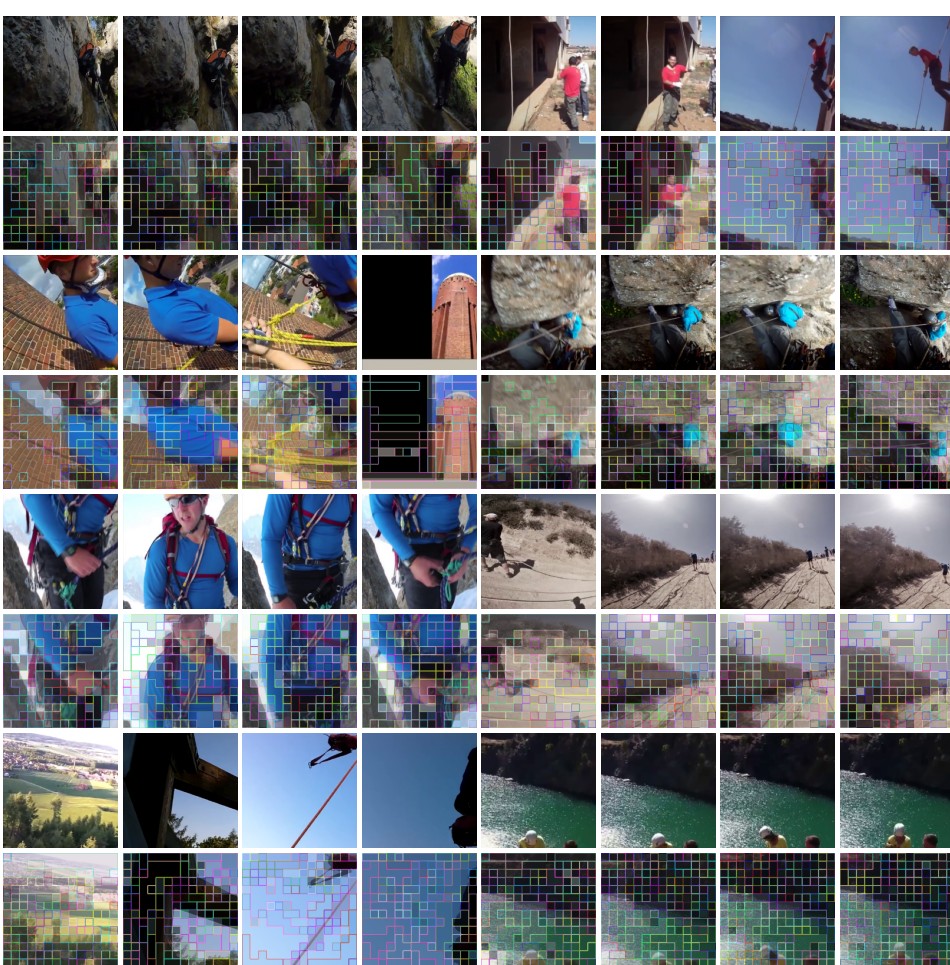

Figure 6: Visualization of R-MeeTo on Kinetics-400 (Carreira & Zisserman, 2017). Tokens belonging to one object across frames are merged into one.

| method | top-1 acc.(%) | | | | FLOPs(G) | | | |
|---|---|---|---|---|---|---|---|---|
| | Vim-Ti | | Vim-S | | Vim-Ti | | Vim-S | |
| Vim (Baseline) (Zhu et al., 2024) | 76.1 | 0.0 | 80.5 | 0.0 | 1.50 | 0.00 | 5.10 | 0.00 |
| Token Recognition (Liang et al., 2022) | 71.3 | 4.8↓ | 74.8 | 5.7↓ | 1.28 | 0.22↓ | 3.57 | 1.53↓ |
| Hidden State Alignment (Zhan et al., 2024) | 75.1 | 1.0↓ | 78.8 | 1.7↓ | 1.29 | 0.21↓ | 3.60 | 1.50↓ |
| R-MeeTo (ours, odd-block) | 75.1 | 1.0↓ | 79.7 | 0.8↓ | 1.27 | 0.23↓ | 3.45 | 1.65↓ |

Table 21: Comparison on the performance of Vim-Ti/S in ImageNet-1K (Deng et al., 2009) classification between different token reduction methods. R-MeeTo (odd-block) achieves higher top-1 accuracy (%) than competing methods while maintaining comparable FLOPs.

| reduction ratio | Vim-Ti | | Vim-S | | Vim-B | |
|---|---|---|---|---|---|---|
| | default (top-$r$) | random-$r$ | default (top-$r$) | random-$r$ | default (top-$r$) | random-$r$ |
| 0.14 | 38.19 | $37.08_{\pm 0.12}$ | 78.52 | $77.96_{\pm 0.09}$ | 78.39 | $78.05_{\pm 0.09}$ |
| 0.28 | 38.41 | $33.45_{\pm 0.19}$ | 76.73 | $75.74_{\pm 0.07}$ | 78.40 | $77.90_{\pm 0.10}$ |
| 0.31 | 35.49 | $30.74_{\pm 0.18}$ | 76.29 | $75.23_{\pm 0.11}$ | 78.28 | $78.24_{\pm 0.07}$ |
| 0.42 | 30.83 | $23.15_{\pm 0.14}$ | 72.86 | $71.53_{\pm 0.12}$ | 77.36 | $77.36_{\pm 0.10}$ |
| 0.54 | 7.44 | $5.10_{\pm 0.10}$ | 60.67 | $56.32_{\pm 0.14}$ | 75.07 | $75.01_{\pm 0.09}$ |

(a) **Top-$r$ v.s. Random-$r$.** Ablate Top-$r$ operation (Line 10 in Algorithm 1) into random selection.

| reduction ratio | Vim-Ti | | Vim-S | | Vim-B | |
|---|---|---|---|---|---|---|
| | default | random pair | default | random pair | default | random pair |
| 0.14 | 38.19 | $35.36_{\pm 0.15}$ | 78.52 | $77.76_{\pm 0.08}$ | 78.39 | $78.27_{\pm 0.08}$ |
| 0.28 | 38.41 | $31.12_{\pm 0.21}$ | 76.73 | $74.06_{\pm 0.12}$ | 78.40 | $77.91_{\pm 0.09}$ |
| 0.31 | 35.49 | $26.91_{\pm 0.17}$ | 76.29 | $73.06_{\pm 0.14}$ | 78.28 | $77.56_{\pm 0.07}$ |
| 0.42 | 30.83 | $14.49_{\pm 0.20}$ | 72.86 | $65.22_{\pm 0.14}$ | 77.36 | $76.16_{\pm 0.09}$ |
| 0.54 | 7.44 | $0.80_{\pm 0.05}$ | 60.67 | $33.80_{\pm 0.19}$ | 75.07 | $72.70_{\pm 0.10}$ |

(b) **Paired merging v.s. Random.** Ablate pairing (Line 3 in Algorithm 4) into random pairing, where we shuffle tokens' indexes between $i$ and $j$ pair.

Table 22: Modules' ablation experiments on default setting. Top-1 accuracy (%) is reported.

