# OpenReview forum: "Faster Vision Mamba is Rebuilt in Minutes via Merged Token Re-training"
_ICLR.cc/2026/Conference — ICLR 2026 Conference Withdrawn Submission_

### Official Review · Reviewer_RNij · 2025-10-24

**Soundness:** 3
**Presentation:** 3
**Contribution:** 2
**Rating:** 4
**Confidence:** 3

**Summary:**

This work introduces token merging to Vision Mamba. The core: (1) due to its sequential state space structure, Mamba is more sensitive to token pruning than Transformers; (2) Token merging preserves more key knowledge than pruning; (3) A short retraining phase can effectively restore performance.

Although the empirical results are convincing, given recent and highly relevant work, novelty is limited.

**Strengths:**

The discussion on token merge and token prunning for Mamba is interesting. This work bring some insights to the efficient vision mamba design.

**Weaknesses:**

The token merging strategy is wildly explore by well-known untrained method on Vision Transformers. Adaptation to Mamba seems like a natural extension. The contribution is limited.

**Questions:**

What`s the difference between a recent works such as "Training-free Token Reduction for Vision Mamba", "Sequential Token Merging: Revisiting Hidden States", which also merge tokens for mamba. Could the authors illustrate original contribution of this work?

---

### Official Review · Reviewer_bp9X · 2025-10-31

**Soundness:** 2
**Presentation:** 2
**Contribution:** 2
**Rating:** 4
**Confidence:** 4

**Summary:**

This paper studies token reduction for Vision Mamba models. The work is practically motivated and proposes a simple general design that consistently recovers most accuracy while cutting FLOPs and improving throughput. The authors propose R-MeeTo with token merging and then perform a short round of re-training. The empirical study includes Vim tiers and VideoMamba. The theoretical section is provided to offer the intuitive lens  but hinges on strong assumptions and interaction-information manipulations that are not fully justified.

**Strengths:**

1. Simple and easy to follow. The merging and re-training mechanisms are not complex and R-MeeTo yields smaller drops at matched or lower FLOPs. This indicates competitive effectiveness at comparable cost. The experiment tables also show that the throughput improves across GPUs with modest accuracy loss. Settings are detailed and this aids replication.
2. Experiments are designed across image and video models and results on Kinetics-400 show consistent behavior and FLOP reductions with small drops. Ablation experiment results are relatively abundant.
3. Visualizations results suggest semantically coherent merges (Figs. 5, 6) improving method intuition to some extent.

**Weaknesses:**

1. Theoretical assumptions and derivations need tighter justification. Assumptions are strong and unvalidated empirically. Consequences for misspecification are not analyzed in Sec. 2.3. This affects technical soundness. The proof of Theorem 1 uses interaction information decompositions with limited rigor. Proposition 2 (no dependency before t) may be an oversimplification for selective SSMs due to the diverse scan order and layer settings of common models such as Vim model. This impacts correctness and clarity.
2. While pruning baselines are compared, there is no direct comparison against prior merging methods adapted, e.g., ToMe-style variants.
3. Some experimental designs are not rigorous. For example, ViT's token interaction method is global attention, so the conclusion that shuffle has no effect may not need to be verified.
4. The writing needs careful proofreading. For example, the abstract sentence "Our key insight is that a quick round of retraining after token merging yeilds robust results across various compression ratios" is missing a period, and the typo "dropa" is in line 73.
5. Hypotheses were listed as a main contribution.

**Questions:**

1. How to accurately define the concept of the term "knowledge" in the paper?
2. For other questions, please refer to the weaknesses part.

---

### Official Review · Reviewer_GQRu · 2025-11-01

**Soundness:** 2
**Presentation:** 2
**Contribution:** 2
**Rating:** 2
**Confidence:** 5

**Summary:**

This paper proposes R-MeeTo (Re-training Merged Token), a simple and effective framework to accelerate Vision Mamba models through token merging followed by lightweight retraining. Unlike pruning-based token reduction methods that cause severe performance drops due to the sequential dependency and information imbalance inherent to Mamba’s state space structure, R-MeeTo merges similar tokens to preserve key knowledge and then retrains the model for a few epochs to recover lost performance. The paper provides both theoretical analyses based on information bottleneck theory and extensive empirical validation. Experiments on ImageNet-1K show that R-MeeTo can recover up to 35.9% accuracy within minutes of retraining, achieving up to 1.5× inference speedup with minimal accuracy degradation across multiple Vision Mamba variants (Vim-Ti/S/B). Additional tests on VideoMamba confirm the method’s generality and scalability across different hardware and data modalities.

**Strengths:**

1. The paper identifies and explains a key inefficiency in token pruning for Mamba models.

2. The paper provides a clear information-theoretic analysis.

3. The propsed model achieves better results than the pruning baseline.

**Weaknesses:**

1. I'm concerning that the paper is comparing with only weak baselines. For example, the Transformer baseline is still DeiT with only 81.8% base-sized model accuracy on ImageNet, which is outdated. A reasonable ViT-Base performance on ImageNet should be over 83.0% (e.g., DeiT-III [1]). Does the conclusion "Mamba is more sensitive that Transformers for pruning" still hold true for DeiT-III?

2. The scalability is unknown. The paper only conduct experiments with base-level models with less than 100M parameters. Does it still work for larger models? I understand that your baseline architectures Vim, VideoMamba, PlainMamba do not have large models, but actually there have already been many recent studies training Mamba at a larger scale like MambaVision [2] and MambaReg [3]. Intuitively, larger models need more input information so I suspect the token pruning/merging approaches cannot work well on them.

3. A clear computation–accuracy trade-off **across all model sizes** is very important for a paper on token pruning or reduction. Although I did not find such a figure or table in the paper, I roughly estimated the trade-off myself. According to your results, after merging 42% of the tokens, Vim-S achieves only 72.9% accuracy. Assuming FLOPs scale linearly with token length (a common assumption for Mamba), the corresponding FLOPs would be around 2.94G, which is still much higher than Vim-Ti’s 1.45G FLOPs, yet the accuracy is significantly lower (76.1% → 72.9%). This raises a question: why not simply reduce the model size (e.g., use Vim-Ti) to achieve better efficiency at similar or even higher accuracy? Such a trade-off appears much more favorable.

4. Besides the lack of experiments on larger model sizes, another limitation of this paper is that it only focuses on a moderate sequence length. If the image resolution is increased or decreased—resulting in a much longer or shorter sequence length—would the conclusions in this paper still hold? It remains unclear whether the proposed method and its observed trends generalize across different sequence lengths.

5.  The formatting of the paper needs significant improvement. Many tables are floating beside the main text, and the excessive use of \vspace makes it difficult to visually distinguish tables from the surrounding paragraphs. It appears that the authors did not carefully adjust the layout—the current manuscript looks as if it was forcefully converted from a two-column template to the single-column ICLR format, resulting in poor readability.

[1] Touvron, Hugo, Matthieu Cord, and Hervé Jégou. "Deit iii: Revenge of the vit." European conference on computer vision. Cham: Springer Nature Switzerland, 2022.

[2] Hatamizadeh A, Kautz J. Mambavision: A hybrid mamba-transformer vision backbone[C]//Proceedings of the Computer Vision and Pattern Recognition Conference. 2025: 25261-25270.

[3] Wang F, Wang J, Ren S, et al. Mamba-Reg: Vision Mamba Also Needs Registers[C]//Proceedings of the Computer Vision and Pattern Recognition Conference. 2025: 14944-14953.

**Questions:**

See Weaknesses.

---

### Official Review · Reviewer_wcPc · 2025-11-04

**Soundness:** 2
**Presentation:** 3
**Contribution:** 2
**Rating:** 4
**Confidence:** 4

**Summary:**

This paper aims to improve the inference efficiency of Vision Mamba (Vim) models. The authors first make two observations: (1) Vision Mamba models are significantly more sensitive to token pruning than standard Vision Transformers (ViTs), leading to severe performance degradation , and (2) Token merging  preserves more information than pruning, but also suffers from performance drops at high compression ratios.

Based on these observations, the paper's central claim is that training-free token reduction is not a good solution for Vision Mamba. The proposed method, R-MeeTo (Re-training Merged Token) , is therefore a simple two-step process: (1) apply token merging, and (2) conduct a very fast (minutes-long) re-training phase to "rebuild the key knowledge" and recover the lost performance. The paper heavily emphasizes the speed of this re-training step as a primary benefit.

**Strengths:**

The paper addresses a timely and practical problem: improving the computational efficiency of Vision Mamba, a promising new architecture.

The initial diagnosis of the problem is correct. The field has indeed observed that Vision Mamba's sequential nature makes it highly sensitive to token reduction methods designed for ViTs.

The proposed solution (merge + fast retrain) is simple and practical, and the reported re-training times (e.g., 5-17 minutes)  are impressive.

**Weaknesses:**

The paper's entire argument and central claim hinge on the assertion that retraining is necessary because "training-free is not a good solution for... Mamba". This claim is unsubstantiated and invalidated by the authors' failure to cite or compare against MTR (Mamba Token Reduction) , a SOTA framework specifically designed for this problem and explicitly advertised as "training-free".

Invalidated Conclusion: Because of this omission, the paper fails to prove its central thesis. It has not shown retraining is necessary; it has only shown that its implementation of merging (which might be suboptimal) can be fixed with retraining. It is possible that the training-free MTR  outperforms R-MeeTo (with retraining), which would render this paper's contribution moot.

**Questions:**

Please see weakness.

---

### Note · Authors · 2025-11-22

**Comment:**

Thank you to all reviewers and area chairs for your time and effort. We will further improve the manuscript accordingly.

Best, Authors of Submission #19976

**Withdrawal Confirmation:**

I have read and agree with the venue's withdrawal policy on behalf of myself and my co-authors.